# Effect of Corn Grinding Methods and Particle Size on the Nutrient Digestibility of Chahua Chickens

**DOI:** 10.3390/ani13142364

**Published:** 2023-07-20

**Authors:** Guoyi Niu, Tingrui Zhang, Shengxiong Cao, Xi Zhang, Linli Tao

**Affiliations:** 1Yunnan Provincial Laboratory of Animal Nutrition and Feed Science, Faculty of Animal Science and Technology, Yunnan Agricultural University, Kunming 650201, China; nguoyi@163.com (G.N.); z943727490@163.com (X.Z.); 2College of Veterinary Medicine, Yunnan Agricultural University, Kunming 650201, China; tingrui_z98@163.com; 3Yun Zhong Mei Agriculture Technology Co., Ltd., Kunming 651701, China; chuyuehan@hotmail.com

**Keywords:** corn, grinding methods, particle size, Chahua chickens, nutrient digestibility

## Abstract

**Simple Summary:**

The methods for improving nutrient digestibility and using feed ingredients have been widely studied. Both hammer and roller mills have their advantages. A combination of both mills can maximize production capacity and cost-effectiveness and achieve an optimal particle size. This study investigates the effect of grinding methods, including roller mill, hammer mill, and two-stage grinding, on the particle size distribution of corn and the effect of corn particle size on the nutrient digestibility of native chickens in Southwest China. Combining roller and hammer mills tends to produce a more uniform particle size. Finely ground corn (between 700 µm and 900 µm) improved the nutrient digestibility of the chickens at week 12, and an increased particle size did not impact the crude protein (CP) and amino acid (AA) digestibility of the chickens at week 19.

**Abstract:**

This study investigates the effect of grinding methods, including roller mill, hammer mill, and two-stage grinding, on the particle size distribution of corn and the effect of corn particle size on the nutrient digestibility of native chickens in Southwest China. The roller mill, hammer mill, and a combination of the hammer mill and roller mill were used to obtain corn with various coarseness. Corn with different coarseness obtained using a combination of the hammer mill and roller mill was fed to Chahua chicken No. 2-type chickens (CHC2s). A total of 192 CHC2s in weeks 12 and 19 were randomly allocated to eight groups in triplicate. The results show that the geometric mean diameter (d_gw_) and the geometric standard deviation (S_gw_) were significantly (*p* < 0.05) affected by the grinding methods. The S_gw_ obtained when using a sieve of 2.0 mm in a hammer mill was lower (*p* < 0.05) than that obtained using a 4.5 mm sieve. Combining the roller mill and hammer mill increased the uniformity of the particle size when grinding coarse particles. For fine particles, the d_gw_ and S_gw_ obtained when using the hammer mill were significantly lower (*p* < 0.05) than those obtained when using the roller mill and two-stage grinding method. Reducing the particle size of the corn (<900 µm) significantly increased the dry matter, crude protein, amino acid digestibility, and apparent metabolizable energy in the chicken in weeks 12 and 19. Fine particles significantly increased the crude protein digestibility of the CHC2s at week 12, while there was no significant effect on the crude protein and amino acid digestibility in the CHC2s at week 19. In conclusion, different grinding methods can affect the particle size distribution. For a coarse particle size, combining the roller mill and hammer mill tends to produce a more uniform particle size. Finely ground corn (between 700 µm and 900 µm) improved the dry matter (DM), apparent metabolizable energy (AME), and crude protein (CP) digestibility of the CHC2s at week 12. An increased particle size did not impact the CP and amino acid (AA) digestibility of the CHC2s at week 19.

## 1. Introduction

The particle size of ingredients is an essential characteristic of a feed as it affects the mixing features, extrusion, pellet quality, growth performance, and gastrointestinal health of animals [1]. Generally, a smaller particle size can increase the surface area available to digestive enzymes and improve an animal’s nutrient digestibility and growth performance [2]. Conversely, coarse particles in poultry diets are important as poultry prefer coarser diets [3]. The feed particle size has been proven to be a factor in developing the gizzards in poultry, and the gizzard weight increases with the consumption of coarsely ground feed [4,5,6]. In addition, ingredients that are too finely ground can result in dust products, feed wastage, energy consumption, and gastric lesions [7]. Therefore, the optimal particle size distribution of ingredients is crucial for animal feed (extrusion, starch gelation) and nutrient digestibility.

Grinding is a standard procedure used to reduce particle size in animal feed. The reduction ratio and grain particle size distribution depend on the milling type and milling variables [8,9]. The hammer mill is the most commonly used mill in the feed industry in China. It can be used to grind various ingredients and is easy to operate. However, the hammer mill can produce ingredients that are too finely ground, which increases energy utilization [10]. Many variables impact the milling output of the hammer mill such as the sieve size, peripheral speed, and hammer number. The feed particle size is mainly controlled by the sieve size in the hammer mill. In contrast, the roller mill is not commonly used as it is unsuitable for milling fibrous materials or hulls. The roller mill reduces the particle size by compression or shear forces based on the roll rotating speed [11]. It can produce more uniform particles and has better energy efficiency than the hammer mill [12]. The speed and gap of the rollers are the variables that affect the particle size distribution of the grain. Both hammer and roller mills have their advantages. Combining both mills can maximize production capacity and cost-effectiveness and achieve an optimal particle size [8,12].

Additionally, the particle size is affected by the grain type and physical characteristics of the grain [13,14]. Corn is the primary material used in poultry feed, and it is highly digestible and a good source of dietary energy [10]. Several studies on the size reduction of corn using different methods and the effects of feed particle size on the nutrient digestibility of poultry have been reported [10,15]. However, the impact of combining the roller mill and hammer mill on the particle size distribution of corn has rarely been reported. This study compares the effect of the roller mill, hammer mill, and a combination of the two mills on the particle size distribution of corn. Chahua chicken No. 2 (CHC2) is a native chicken used for meat and egg production in the Southwest District and is generally marketed at 120 days of age with an average weight of 2 kg. This chicken is broadly classified as a slow-growing yellow-feathered breed. This breed has a similar growth rate to the Beijing You chicken and the Lingnan breed [16,17]. Consumers highly desire the distinct flavor of the meat. Therefore, this research aims to evaluate the effect of different grinding methods (roller mill, hammer mill, and two-stage grinding) on the particle size distribution of corn, and the impact of varying coarseness of corn on dry matter (DM), crude protein (CP), amino acid (AA) digestibility, and apparent metabolizable energy (AME) in CHC2s in Southwest China.

## 2. Materials and Methods

All animal protocols were conducted with the approval of the Animal Care and Use Committee of Yunnan Agricultural University, China (Approval No. 202103032).

### 2.1. Feed Ingredients

The corn was obtained from Kunming Yunling Broad breed poultry Feed Co., Ltd. (Kunming, China). The corn used during this study was the same batch. The proximate nutrient composition of the corn is presented in Table 1.

### 2.2. Grinding Methods

In this experiment, a roller mill, a hammer mill, and a combination of a hammer and a roller mill were employed to obtain optimal coarseness according to previous results. This study was designed to investigate the effect of a roller mill, a hammer mill, and a combination of a hammer mill and a roller mill on corn particle size distribution (600–1180 µm). The conditions of the roller mill, the hammer mill, and the two-stage grinding are shown in Table 2. The roller mill (MengNan, Zhengzhou, China) was fitted with two rollers, with a differential speed of 1:2.5 for this pair of rollers. The speed of the fast roller was 6 and 8 m/s with the gap between the two rollers being 0.4, 0.6, 0.8, and 1.0 mm, respectively. Different peripheral speeds (55, 70, 85 m/s), hammer tip to sieve clearance distances (12, 9, 6 mm), hammer thicknesses (2, 4 mm), hammer numbers (8, 12 hammers), and sieve opening diameters (2.0 and 4.5 mm) were selected in the hammer mill (SanZhang, Nanyang, China). The three grinding methods (roller mill, hammer mill, and two-stage grinding) were employed under M1, M2, M3, and M4 conditions. Twelve different particle sizes of corn were obtained and measured. One kilo of ground corn per particle size was collected in triplicate.

### 2.3. Measurement of Particle Size

The particle size of corn was determined in triplicate per group using dynamic image analysis (BT-2900; Bettersize, Dandong, China). The particle size was distributed as follows: 53; 75; 106; 150; 212; 300; 425; 600; 850; 1180; 1700; 2360; 3350; and 4750 µm (GB 6971-86). The geometric mean diameter (d_gw_) and geometric standard deviation (S_gw_) were calculated according to the equations described by ASAE standard [18], which are as follows.
$$d_{gw}={log}^{-1}\left[\frac{{\sum W}_{i}\log\;d_{i}}{W}\right]$$
$$S_{gw}={log}^{-1}{\left[\frac{{{\sum W}_{i}\left(\log\;\bar{d_{i}}-\log\;d_{gw}\right)}^{2}}{W}\right]}^{\frac{1}{2}}$$
where d_i_ (µm) and d_i+1_ (µm) are the size of sieve openings of ith and (i + 1)th sieve, $\bar{d_{i}}=\sqrt{d_{i}\times d_{i+1}}$, W_i_ is the proportion on ith sieve, W = 100.

### 2.4. Experiment Design and Chickens

This experiment was conducted at a rearing farm at Yunnan Agricultural University. In this experiment, we fed corn of four different particle sizes using the two-stage grinding method to CHC2s in week 12 (growing period) and 19 (market age). A total of 192 CHC2s in weeks 12 and 19 were randomly allocated to eight groups, in triplicate for each group, with eight chickens per replicate (four males and four females), and the average weights of CHC2s at weeks 12 and 19 were 1120.42 g and 1625.73 g, respectively. The chickens were raised in four-story cages, and the replicate cages were evenly distributed in the chicken house. The chickens were immunized according to routine immunization procedures. The stocking density was approximately 5 to 6 birds/m^2^. The room temperature was maintained at 33 to 34 °C for the first week, and then the temperature was gradually decreased by two degrees every week until it reached 21 °C. For the first week, 24 h of light was provided, and then it was reduced by 2 h every week and maintained at 18 h until the end of the trial. The relative humidity was maintained at 55% to 65%. Feed and water were provided ad libitum throughout the trial. The composition and nutrient levels of the diets at the ages of 0–11 weeks and 13–18 weeks are shown in Table 3.

### 2.5. Determination of Nutrient Digestibility

The AME was determined using the total excreta collection procedure [19]. During each week, chickens were fed with corn for seven days, with the first three days serving as an adaptation period. The feed intake (FI) and total excreta output for each replicate cage were recorded over the last four consecutive days. The total excreta were collected daily and mixed with 10% HCl (0.1 mL g^−1^). Feed spillage was measured from the under-cage collection tray and deducted from feed intake. Feathers and down were removed from the collected excreta [20]. Each sample was dried, ground to pass through a 0.5 mm-sieve, and stored in a plastic container at 4 °C pending analysis. The DM, gross energy (GE), crude protein (CP), and amino acid (AA) were measured in the diets and excreta.

DM was determined according to the Association of Official Analytical Chemists [21]. GE was measured by means of a microcomputer employing a completely automatic calorimetry meter (ZDHW-6, Huatai Instrument Measuring Appliance Co., Ltd., Zhuozhou, China). CP was analyzed using a Kjeldahl nitrogen tester (SKD-1000, Peiou Analytical Instruments Co., Ltd. Shanghai, China). The AA was determined through high-performance liquid chromatography (HPLC, LC-20A, Shimadzu CO., Ltd., Tokyo, Japan).

Apparent metabolizable energy (AME), digestibility of DM, CP, and AA were calculated according to the equation.
AME (MJ kg^−1^) = (FI × GE_diet_ − excreta output × GE_excreta_)/FI
Nutrient digestibility (%) = (FI × N_diet_ − excreta output × N_excreta_)/(FI × N_diet_) × 100%
where FI is feed intake, GE_diet_ is gross energy of the diet, GE_excreta_ is gross energy of the excreta, N_diet_ is the nutrient content in diet, N_excreta_ is the nutrient content in excreta.

### 2.6. Statistical Analysis

All of the data were analyzed via one-way ANOVA using SPSS software version 23 (IBM Corporation, New York, NY, USA). Duncan’s test was used to compare data among treatments. Data were presented as mean ± standard deviation (SD), and the significance of effects was declared at *p* < 0.05.

## 3. Results

### 3.1. Effect of Grinding Method on the Particle Size Distribution

The d_gw_ and S_gw_ were significantly affected by grinding methods (Table 4). For fine particle size (groups M1 and M2), the d_gw_ and S_gw_ produced using a hammer mill were significantly lower than those produced using a roller mill and the two-stage grinding method (*p* < 0.05). For group M1, combining a roller mill and a hammer mill significantly decreased the d_gw_ compared to the roller mill, while no significant effect on S_gw_ was observed between these two methods. Using the two-stage grinding method significantly decreased the S_gw_ compared to the roller mill in group M2, while there was no significant difference in d_gw_ between these two methods. For coarse particle sizes (groups M3 and M4), the d_gw_ produced using a hammer mill was significantly lower than that produced using a roller mill, while using a hammer mill significantly increased the S_gw_ compared to the two-stage grinding method (*p* < 0.05). Using the two-stage grinding method significantly decreased the d_gw_ compared to the roller mill in group M3, and no significant difference in S_gw_ was observed between the two methods. Using the two-stage grinding method significantly decreased (*p* < 0.05) the S_gw_ compared to using the hammer mill or the roller mill in group M4.

### 3.2. Effect of Particle Size on the Nutrient Digestibility

The feed intake, excreta content, and nutrient composition of the excreta of the CHC2s in week 12 were shown in Table 5. The chickens fed with corn with coarse particle sizes displayed a significantly increased excreta content and a decreased CP content of excreta (*p* < 0.05). The FI and amino acid content of excreta were not significantly affected by corn particle size. The particle size demonstrated a significant effect (*p* < 0.05) on the DM, CP and AA digestibility and AME of the CHC2s in week 12 (Table 6). The DM and CP digestibility and the AME of the CHC2s that were fed the M4 diet were significantly lower than those of chickens fed the M1 and M2 diets (*p* < 0.05). The Ser, Met, and Cys digestibility of the CHC2s fed the M3 and M4 diets was significantly lower than that of the chickens fed the M1 diet (*p* < 0.05). No significant effect on the digestibility of the other amino acids was observed among these diets. The Tyr, Val, Ile, Leu, Phe, and Lys digestibility of the CHC2s gradually decreased with the increase in corn particle size.

The feed intake, excreta content, and nutrient composition of the excreta of the CHC2s in week 19 are shown in Table 7. The chickens fed corn with coarse particle size significantly increased GE of excreta (*p* < 0.05). The CP content of excreta of the CHC2s fed the M4 diet was significantly lower than that of the chickens fed the fine particle size corn. Corn particle size did not significantly affect the FI, excreta content, and amino acid content of excreta. The particle size demonstrated a significant effect (*p* < 0.05) on the DM digestibility and the AME of the CHC2s in week 19 (Table 8). The DM digestibility and the AME of CHC2s fed the M4 diet were significantly lower than those of chickens fed the M1and M2 diets (*p* < 0.05). No significant difference in the CP and AA digestibility was observed in the CHC2s in week 19. The amino acid digestibility of CHC2s fed the corn with fine particle size was higher than that of the chickens fed the corn with coarse particle size except for Gly, Pro, and Met.

## 4. Discussions

### 4.1. Effect of Grinding Method on the Particle Size Distribution

This study investigated the effect of different grinding methods, including a roller mill, a hammer mill, and two-stage grinding on the particle size distribution of corn.

Many factors could impact the efficiency of the size reduction ratio of mill devices [7]. A roller mill comprises one or more pairs of rollers. The particle size of corn is reduced by a constant compression force as they pass between the rotating rollers. The smaller the roller gap, the higher the pressure of the roller. The S_gw_ indicates particle size uniformity. The lower the S_gw_, the more uniform the particles are. Consistent with the results obtained from Gebhardt [22], our study showed that decreasing the roller gap could increase the particle size uniformity.

A hammer mill comprises a set of hammers moving at high speed in the grinding chamber, which reduces the size of the corn until the particles can pass through designated sieves [7,8]. The particle size and its uniformity are controlled by the sieve size and the hammer speed [23]. The present results show that the d_gw_ and S_gw_ are mainly affected by the sieve size in a hammer mill. Decreasing the sieve size in a hammer mill could increase the particle size uniformity. Similarly, the research observed by Vukmirović [12] indicated that the geometric standard deviations were 2.66, 2.82, and 2.69 when using a hammer mill with sieve openings diameters of 3, 6, and 9 mm, respectively. Dey [24] reported that the reduction ratio increases with increased rotor speed. However, several studies showed that higher energy consumption was obtained with decreasing sieve size [8,12,25,26]. Thomas [8] reported that roller mills are the most energy-efficient coarse grinding devices. Moreover, coarsely rolled corn was more uniform than hammer milled corn [12].

The mechanical differences make it difficult to compare the efficiency of roller mills and hammer mills. We compared the effect of grinding methods on the particle size distribution when the particle size ranged from 600 to 1180 µm. Both roller mills and hammer mills have advantages and drawbacks [1]. Two-stage grinding combines a hammer mill with a roller mill. Normally, two-stage grinding often contains a sieving step after first grinding to sieve out all particles of an undesired size. The current results show that using the two-stage grinding method could increase the particle size uniformity for coarse particles. The hammer mill produced a more uniform particle size for finely ground corn, but it can produce too finely ground ingredients and increase energy utilization [10]. Similarly, Lucht [27] revealed that combined-stage grinding with a hammer mill and a crushing roller mill resulted in both a reduced fines content and a high percentage within the medium range of 0.5 to 1.6 mm. Therefore, these results indicate that the two-stage grinding method combines the advantages of hammer mill and roller mill, which may provide a suitable particle size at the lowest cost. It is important to combine these devices to ensure an efficient milling operation and to obtain specific grinding objectives.

### 4.2. Effect of Particle Size on the Nutrient Digestibility

Particle size has been proven to be an important factor in gizzard development and nutrient utilization in poultry. Particle size reduction could improve nutrients’ digestibility by increasing the surface area available to digestive enzymes [2]. The present research showed that decreasing particle size (600–900 μm) of corn significantly increased the AME of the CHC2s at weeks 12 and 19, which is in accordance with previous results [7]. Kilburn and Edwards [28] reported that finely grinding maize improved the metabolizable energy values of diets.

In contrast, several studies reported that coarsely grinding corn increased nutrient digestibility and growth performance [29]. Chickens are shown to select the larger feed particles. Fines or dust negatively influence feed intake, as fines are difficult to pick up using the bird’s beak [30,31]. In this study, the CP and AA digestibility of the CHC2s of market age was not affected by particle size. Similarly, Mtei et al. [32] indicated that increasing the corn particle size from 490 to 900 μm led to a higher coefficient of apparent ideal digestibility of DM and GE. The hens fed with coarsely ground diets had a higher feed conversion ratio than hens fed with finely ground diets [33]. Therefore, using the optimal particle size is crucial for poultry of different ages to maximize growth performance. Using the most suitable milling devices or operations not only decreases energy consumption, but also increases the nutrient digestibility of poultry. The effect of corn particle size on nutrient digestibility displayed a variety of results in various studies. Bird type and the physical properties of the grain may contribute to these varieties [5,13,32]. Obviously, more uniform particles could reduce the time spent searching for larger particles, and thus improve the growth performance of poultry. Thus, the effect of particle size uniformity for corn on nutrient digestibility should be investigated in further research.

## 5. Conclusions

It can be concluded that grinding methods can affect the particle size distribution. For a coarse particle size, combining the roller mill and hammer mill tends to produce a more uniform particle size. Finely ground corn (between 700 µm and 900 µm) improved the DM, AME and CP digestibility of the CHC2s at week 12. An increased particle size did not impact the CP and AA digestibility of the CHC2s at market age.

## Figures and Tables

**Table 1 animals-13-02364-t001:** The nutrient composition of corn (DM basis).

	Corn
DM (%)	86.65
GE (MJ/kg)	17.29
CP (%)	10.83
Asp (%)	0.70
Glu (%)	2.19
Ser (%)	0.49
Gly (%)	0.33
His (%)	0.26
Arg (%)	0.53
Thr (%)	0.38
Ala (%)	0.75
Pro (%)	0.78
Tyr (%)	0.21
Val (%)	0.36
Met (%)	0.03
Cys (%)	0.17
Ile (%)	0.31
Leu (%)	1.06
Phe (%)	0.39
Lys (%)	0.27
TAA (%)	9.21

**Table 2 animals-13-02364-t002:** The conditions of the roller mill, the hammer mill, and the two-stage grinding.

Group	Roller Mill		Hammer Mill				
	Roller Gap (mm)	Roller Speed (m/s)	Peripheral Speed (m/s)	Hammer Tip to Sieve Clearance (mm)	Hammer Thickness (mm)	Hammer Numbers	Sieve Size (mm)
M1	0.4	6	85	6	4	12	2.0
M2	0.6	8	70	9	4	8	2.0
M3	0.8	6	55	12	2	8	4.5
M4	1.0	8	55	12	2	8	4.5

**Table 3 animals-13-02364-t003:** Composition and nutrient levels of the diets.

Ingredients	0–6 w	7–11 w	13–18 w
Corn	55.00	60.00	65.00
Soybean meal	25.00	15.00	15.00
Corn gluten meal	5.59	8.96	7.31
Wheat bran	6.83	7.04	4.51
Soybean oil	3.13	4.25	3.49
Limestone	1.00	1.20	1.08
Dicalcium phosphate	1.50	1.50	1.49
L–Lysine HCl	0.44	0.50	0.55
DL-methionine	0.26	0.30	0.32
Sodium chloride	0.25	0.25	0.25
Premix ^1^	1.00	1.00	1.00
Total	100	100	100
Nutritional levels ^2^			
Metabolizable energy (MJ/kg)	12.69	13.52	13.55
Crude Protein (%)	20.10	18.52	17.68
Methionine (%)	0.62	0.51	0.52
Lysine (%)	1.08	1.09	1.09
Calcium (%)	0.85	0.91	0.85
Total phosphorus (%)	0.66	0.63	0.61
Available phosphorus (%)	0.47	0.36	0.35

^1^ The premix provided per kilogram of diets aged 0–11 weeks: vitamin A 5000 IU; vitamin D3 1000 IU; vitamin E 10 IU; vitamin K 0.5 mg; thiamine 1.8 mg; riboflavin 3.6 mg; niacin 35 mg; pantothenic acid 20 mg; pyridoxine 3.5 mg; biotin 0.15 mg; folic acid 0.6 mg; cobalamin 0.01 mg; choline chloride 1000 mg; copper 8 mg; iron 80 mg; zinc 90 mg; manganese 80 mg; iodine 0.25 mg; selenium 0.15 mg. The premix provided per kilogram of diets aged 13–18 weeks: vitamin A 5000 IU; vitamin D3 1500 IU; vitamin E 15 IU; vitamin K 0.5 mg; thiamine 2.3 mg; riboflavin 4.0 mg; niacin 35 mg; pantothenic acid 20 mg; pyridoxine 4.0 mg; biotin 0.2 mg; folic acid 0.6 mg; cobalamin 0.01 mg; copper 8 mg; iron 100 mg; zinc 100 mg; manganese 100 mg; iodine 0.25 mg; selenium 0.25 mg. ^2^ Metabolizable energy, crude protein, methionine, and lysine are measured values, the rest of the nutritional levels are calculated values.

**Table 4 animals-13-02364-t004:** Effect of the grinding type on corn particle size distribution.

	Group	Grinding Type		
		Roller Mill	Hammer Mill	Two-Stage Grinding
d_gw_ (μm)	M1	821.06 ± 31.26 ^a^	668.48 ± 3.79 ^c^	757.65 ± 38.37 ^b^
	M2	963.59 ± 92.87 ^a^	724.82 ± 1.80 ^b^	907.72 ± 91.66 ^a^
	M3	1255.08 ± 102.44 ^a^	950.95 ± 13.23 ^b^	1081.86 ± 93.98 ^b^
	M4	1112.28 ± 48.27 ^a^	950.95 ± 13.23 ^b^	1107.65 ± 38.28 ^a^
S_gw_	M1	1.82 ± 0.04 ^a^	1.76 ± 0.02 ^b^	1.80 ± 0.03 ^a^
	M2	1.99 ± 0.06 ^a^	1.78 ± 0.01 ^c^	1.92 ± 0.01 ^b^
	M3	2.04 ± 0.01 ^b^	2.12 ± 0.02 ^a^	2.01 ± 0.05 ^b^
	M4	2.10 ± 0.02 ^a^	2.12 ± 0.02 ^a^	2.05 ± 0.02 ^b^

Values with different letters in the same row are significantly different (*p* < 0.05).

**Table 5 animals-13-02364-t005:** The feed intake, excreta content, and nutrient composition of the excreta of the CHC2s in week 12.

	M1	M2	M3	M4
FI (g)	745.41 ± 4.43	743.10 ± 1.69	748.26 ± 2.29	746.02 ± 4.55
Excreta (g)	125.62 ± 2.06 ^b^	131.61 ± 5.35 ^b^	144.64 ± 4.01 ^a^	150.97 ± 2.28 ^a^
GE (MJ/kg)	16.96 ± 0.04 ^b^	17.61 ± 0.26 ^a^	17.70 ± 0.39 ^a^	17.44 ± 0.14 ^a^
CP (%)	33.71 ± 0.82 ^a^	31.95 ± 1.47 ^a^	30.01 ± 0.81 ^b^	28.99 ± 0.76 ^b^
Asp (%)	1.13 ± 0.07	1.12 ± 0.08	1.13 ± 0.04	1.04 ± 0.11
Glu (%)	1.82 ± 0.27	1.73 ± 0.16	1.74 ± 0.09	1.59 ± 0.18
Ser (%)	0.78 ± 0.02	0.88 ± 0.10	0.81 ± 0.05	0.76 ± 0.05
Gly (%)	0.92 ± 0.06	0.88 ± 0.06	0.82 ± 0.02	0.83 ± 0.06
His (%)	0.24 ± 0.03	0.24 ± 0.02	0.26 ± 0.04	0.24 ± 0.02
Arg (%)	0.82 ± 0.06	0.74 ± 0.02	0.79 ± 0.05	0.72 ± 0.07
Thr (%)	0.70 ± 0.05	0.68 ± 0.05	0.69 ± 0.01	0.64 ± 0.05
Ala (%)	0.73 ± 0.06	0.72 ± 0.05	0.73 ± 0.04	0.68 ± 0.07
Pro (%)	0.90 ± 0.07	0.88 ± 0.06	0.79 ± 0.05	0.80 ± 0.05
Tyr (%)	0.30 ± 0.03	0.29 ± 0.04	0.28 ± 0.02	0.29 ± 0.05
Val (%)	0.59 ± 0.05	0.58 ± 0.05	0.55 ± 0.01	0.53 ± 0.06
Met (%)	0.04 ± 0.00	0.04 ± 0.00	0.04 ± 0.00	0.04 ± 0.00
Cys (%)	0.11 ± 0.00	0.11 ± 0.00	0.10 ± 0.00	0.11 ± 0.00
Ile (%)	0.49 ± 0.04	0.48 ± 0.05	0.45 ± 0.01	0.44 ± 0.05
Leu (%)	0.81 ± 0.07	0.80 ± 0.08	0.76 ± 0.04	0.74 ± 0.09
Phe (%)	0.47 ± 0.04	0.46 ± 0.05	0.43 ± 0.01	0.42 ± 0.05
Lys (%)	0.67 ± 0.08	0.64 ± 0.06	0.65 ± 0.04	0.63 ± 0.08
TAA (%)	11.53 ± 0.89	11.27 ± 0.90	11.03 ± 0.41	10.51 ± 1.04

Values with different letters in the same row are significantly different (*p* < 0.05).

**Table 6 animals-13-02364-t006:** The effect of corn particle size on the digestibility of dry matter (DM, %), apparent metabolizable energy (AME, MJ/kg), crude protein (CP, %) and amino acid (AA, %) of the CHC2s in week 12 (DM basis).

	M1	M2	M3	M4
DM	83.15 ± 0.35 ^a^	82.29 ± 0.75 ^a^	80.67 ± 0.48 ^b^	79.76 ± 0.27 ^b^
AME	14.43 ± 0.07 ^a^	14.17 ± 0.12 ^b^	13.87 ± 0.01 ^c^	13.76 ± 0.02 ^c^
CP	47.57 ± 0.97 ^a^	47.83 ± 0.18 ^a^	46.48 ± 0.82 ^ab^	45.85 ± 0.75 ^b^
Asp	72.72 ± 1.88	71.61 ± 2.5	68.71 ± 0.47	69.67 ± 3.18
Glu	85.96 ± 2.22	86.04 ± 1.41	84.62 ± 0.59	85.33 ± 1.72
Ser	73.14 ± 0.58 ^a^	68.33 ± 3.37 ^b^	68.06 ± 2.49 ^b^	68.36 ± 2.23 ^b^
Gly	52.48 ± 4.09	52.71 ± 3.18	51.40 ± 2.35	48.78 ± 3.93
His	84.31 ± 1.83	83.62 ± 2.17	80.52 ± 2.90	81.08 ± 1.68
Arg	74.13 ± 2.38	75.48 ± 1.30	71.63 ± 1.78	72.74 ± 2.63
Thr	69.17 ± 2.74	68.82 ± 2.50	65.35 ± 0.34	66.54 ± 2.86
Ala	83.50 ± 1.49	82.87 ± 1.50	81.18 ± 0.70	81.62 ± 1.96
Pro	80.47 ± 1.92	79.95 ± 1.43	80.31 ± 1.37	79.20 ± 1.47
Tyr	76.37 ± 2.39	75.54 ± 3.63	73.99 ± 1.41	72.51 ± 4.83
Val	72.13 ± 2.58	71.14 ± 2.82	70.43 ± 1.00	70.06 ± 3.24
Met	79.99 ± 0.99 ^a^	79.49 ± 1.53 ^a^	76.35 ± 0.97 ^b^	77.30 ± 1.67 ^b^
Cys	89.51 ± 0.21 ^a^	88.55 ± 0.93 ^ab^	88.43 ± 0.30 ^b^	86.89 ± 0.30 ^c^
Ile	73.39 ± 2.29	72.21 ± 2.99	71.99 ± 0.77	71.08 ± 3.17
Leu	87.02 ± 1.31	86.55 ± 1.49	86.05 ± 0.75	85.84 ± 1.68
Phe	79.79 ± 2.00	79.15 ± 2.18	78.73 ± 0.53	78.02 ± 2.45
Lys	57.99 ± 5.36	57.62 ± 4.11	53.45 ± 1.57	52.57 ± 6.23
TAA	78.90 ± 1.87	78.32 ± 1.92	76.86 ± 0.72	76.91 ± 2.33

Values with different letters in the same row are significantly different (*p* < 0.05).

**Table 7 animals-13-02364-t007:** The feed intake, excreta content, and nutrient composition of the excreta of the CHC2s in week 19.

	M1	M2	M3	M4
FI (g)	864.30 ± 75.87	836.51 ± 72.95	897.88 ± 52.61	817.07 ± 67.01
Excreta (g)	139.48 ± 6.75	140.32 ± 8.45	153.12 ± 7.96	148.10 ± 8.12
GE (MJ/kg)	17.33 ± 0.55 ^b^	17.04 ± 0.75 ^b^	18.83 ± 0.02 ^a^	19.05 ± 0.67 ^a^
CP (%)	34.20 ± 1.53 ^a^	32.88 ± 1.17 ^a^	32.57 ± 0.56 ^ab^	30.71 ± 1.06 ^b^
Asp (%)	0.86 ± 0.05	0.88 ± 0.08	0.90 ± 0.03	0.90 ± 0.14
Glu (%)	1.29 ± 0.05	1.29 ± 0.11	1.32 ± 0.04	1.33 ± 0.21
Ser (%)	0.60 ± 0.05	0.54 ± 0.06	0.60 ± 0.03	0.57 ± 0.14
Gly (%)	0.69 ± 0.03	0.65 ± 0.05	0.64 ± 0.03	0.66 ± 0.09
His (%)	0.21 ± 0.01	0.21 ± 0.03	0.23 ± 0.01	0.22 ± 0.03
Arg (%)	0.64 ± 0.05	0.61 ± 0.04	0.69 ± 0.07	0.68 ± 0.11
Thr (%)	0.54 ± 0.02	0.52 ± 0.05	0.56 ± 0.01	0.53 ± 0.10
Ala (%)	0.53 ± 0.03	0.53 ± 0.04	0.55 ± 0.02	0.56 ± 0.07
Pro (%)	0.60 ± 0.04	0.57 ± 0.03	0.59 ± 0.03	0.60 ± 0.11
Tyr (%)	0.20 ± 0.01	0.20 ± 0.02	0.20 ± 0.00	0.20 ± 0.02
Val (%)	0.39 ± 0.03	0.37 ± 0.03	0.38 ± 0.02	0.39 ± 0.07
Met (%)	0.03 ± 0.00	0.03 ± 0.00	0.03 ± 0.00	0.03 ± 0.01
Cys (%)	0.09 ± 0.01	0.09 ± 0.00	0.08 ± 0.00	0.09 ± 0.01
Ile (%)	0.32 ± 0.03	0.31 ± 0.03	0.31 ± 0.01	0.33 ± 0.05
Leu (%)	0.52 ± 0.03	0.51 ± 0.05	0.53 ± 0.02	0.54 ± 0.09
Phe (%)	0.30 ± 0.02	0.29 ± 0.02	0.30 ± 0.01	0.31 ± 0.05
Lys (%)	0.47 ± 0.02	0.48 ± 0.05	0.50 ± 0.02	0.49 ± 0.06
TAA (%)	8.28 ± 0.44	8.09 ± 0.63	8.41 ± 0.30	8.44 ± 1.35

Values with different letters in the same row are significantly different (*p* < 0.05).

**Table 8 animals-13-02364-t008:** The effect of corn particle size on the digestibility of dry matter (DM, %), apparent metabolizable energy (AME, MJ/kg, DM basis), crude protein (CP, %) and amino acid (AA, %) of the CHC2s in week 19 (DM basis).

	M1	M2	M3	M4
DM	83.83 ± 0.61 ^a^	83.20 ± 0.47 ^a^	82.94 ± 0.12 ^a^	81.85 ± 0.48 ^b^
AME	14.49 ± 0.02 ^a^	14.43 ± 0.05 ^a^	14.08 ± 0.03 ^b^	13.83 ± 0.04 ^c^
CP	49.00 ± 0.57	49.05 ± 0.52	48.72 ± 0.52	48.58 ± 0.78
Asp	79.94 ± 1.36	78.85 ± 2.43	77.99 ± 0.56	76.45 ± 3.98
Glu	90.44 ± 0.64	90.11 ± 1.07	89.70 ± 0.26	88.91 ± 1.94
Ser	80.28 ± 2.34	81.37 ± 2.47	79.12 ± 0.89	78.74 ± 5.71
Gly	65.77 ± 2.60	66.63 ± 2.31	66.70 ± 1.41	63.42 ± 5.47
His	87.14 ± 0.62	86.45 ± 2.11	84.98 ± 0.33	84.41 ± 2.45
Arg	80.70 ± 1.51	80.76 ± 1.67	77.88 ± 2.06	76.89 ± 3.94
Thr	77.25 ± 1.62	77.27 ± 2.73	75.35 ± 0.16	74.82 ± 5.28
Ala	88.54 ± 0.87	88.00 ± 1.09	87.54 ± 0.34	86.46 ± 2.06
Pro	87.54 ± 1.09	87.76 ± 1.01	87.13 ± 0.55	86.01 ± 2.84
Tyr	84.87 ± 1.39	84.33 ± 1.84	84.19 ± 0.29	82.95 ± 2.51
Val	82.59 ± 1.65	82.42 ± 1.89	81.81 ± 0.62	80.02 ± 3.86
Met	84.27 ± 1.65	82.91 ± 2.07	84.35 ± 0.44	83.07 ± 5.44
Cys	91.60 ± 0.96	91.11 ± 0.29	91.64 ± 0.14	90.20 ± 1.45
Ile	83.44 ± 1.69	83.06 ± 1.71	82.69 ± 0.48	80.84 ± 3.39
Leu	92.01 ± 0.67	91.86 ± 0.89	91.47 ± 0.29	90.72 ± 1.71
Phe	87.50 ± 1.15	87.27 ± 1.29	86.70 ± 0.40	85.53 ± 2.59
Lys	71.54 ± 1.14	69.97 ± 3.55	68.45 ± 0.81	66.86 ± 4.79
TAA	85.45 ± 1.07	85.22 ± 1.46	84.43 ± 0.45	83.33 ± 2.98

Values with different letters in the same row are significantly different (*p* < 0.05).

## Data Availability

Not applicable.

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
