# Peer review of "Effect of Corn Grinding Methods and Particle Size on the Nutrient Digestibility of Chahua Chickens"

_animals, 2023, doi:10.3390/ani13142364_

Round 1
Reviewer 1 Report
Summary
This study examined the grinding of corn using hammer mill, roller mill or a combination of the two methods. This study is important because each method has its advantages over the other in 1. The cost involved such as the energy cost to operate 2. The national benefit on the chicken that consumes it. One of the main contributions is testing these particle sizes on the Chahua chicken No.2 breed. This is because there can be a breed effect when testing different feed particle sizes. This study will also benefit feed manufacturers that have access to both roller and hammer mills to be able to use both.
General comments:
The greatest challenge is that, not every feed mill has access to both roller and hammer mills. They are expensive and so people normally have only one type. The energy used to operate both will be very expensive.
It would be good to show the diet formulation with all the ingredients and feed additives used in the feed because that can also cause some differences in the nutrients.
Were there any differences in production parameters such as weight gain, feed efficiency or egg production in layers?
It would be great to add the economics or how much it would cost to grind corn using each method or both and that is important in deciding on which method to use.
Suggestions:
Effect of corn grinding methods and particle size on nutrient digestibility in native chicken
Line 70: Depends
Line 76: Reduce
Line 78: Have
This paper is well written.
Reviewer 2 Report
Manuscript is interesting and rise important insights regarding the production of commercial chicken fed.
Yet, it suffers from some minor problems; most of them can easily be solved by revising the manuscript.
Material and methods:
· table #1, should be placed post section 2.1, which refers to it and not post the introduction before it is mentioned.
· Line 102: table 2 seems to be out of place. There is no nearby reference in the text to this table. It should be relocated post section 2.2 in the material and methods chapter.
· lines 108-110, are superscript on table #2 data.
· Section 2.4: please add details regarding the chickens' husbandry (water, light regime, temperature, humidity and density).
Results:
· Table #3 should be placed after section 3.1 which refers to it and after the entire results text.
· Lines 202-203, compare between two-stage and roller mill grinding for solely the M1 group. This is a very partial depiction of the obtained observation. Comparison between two stage and hammer grinding for M1 group and parallel comparisons for the M2 group should be added.
· Lines 204-205: authors argue "For coarse particle size (M3 and M4 group), the dgw using two-stage grinding method was significantly lower than that using roller mill (P < 0.05)". That is not supported in the raw results: roller mill grinding is not significant from M4 two-stage grinding group for dgw. Please revise.
· Similarly to the comment regarding the fine particles size, comparisons between treatments and groups are missing. Please revise.
· Lines 207-208: authors argue "Moreover, two-stage grinding method significantly decreased (P < 0.05) the Sgw compared to use hammer mill or roller mill when grinding coarse particles (M4 group)". This is not supported by the M3 group which was treated the same as M4 group. Please revise.
· Section 3.2:
o Authors argue that particles size shows significant effect on CP digestibility and that it is significantly decrease with the increase in particles size. This argument is only partially supported by the observation (group M3). That is also the case regarding AA digestibility in general (only a few AA, supports this argument). Please revise.
o Lines 215-216, compare only M3, M4 and M1 groups for only Ser digestibility. Those are only partial comparisons. Comparisons regarding different AA digestibility should be added.
o Line 217-218: the argument does not support the observation regarding DM (group M3). Please revise.
o Lines 220-221: this argument is partially correct due to effect on AA digestibility and partial effect on the DM digestibility. Please revise.
Discussion and conclusions:
· Lines 264-265: according to current observations, the use of 2-stage grinding increase particles uniformity only (and not especially) for coarse particles. Moreover, it seems that for fine particles, better uniformity is achieved using hammer grinding rather than 2-stage grinding. Please revise.
· Lines 276-277: this argument is partially correct regarding DM at week 19 (M3 group). Please revise.
· Line 298: "Coarsely rolled corn is more uniform than hammer milled corn". According to the observations, it's correct only for one out of the two tested groups (M4 group does not show any difference between these two grinding methods, regarding uniformity). Please revise.
· Lines 298-299: "Grinding coarse particles 298 using a two-stage mill could increase the particle size uniformity". According to the observations, it's correct only when comparing to hammer grinding and partially when comparing to roller grinding (M3 group does not show any difference between these two grinding methods, regarding uniformity). Please revise.
· Lines 299-300: "Reducing the particle size of corn (between 700μm and 900μm) improves the nutrient digestibility in chicken". This is also partially correct; it is depended on the chicken age and the tested digestibility parameter. Please revise.
Introduction:
· Line 57: there are extra commas before and after the word "and". Please revise the sentence.
· Lines 61-62: the sentence: "… to be a factor in the development of gizzard, and the gizzard weight increased with coarse ground feed" is not clear. Please revise.
· Line 64: there is an extra comma before the word "and" – please revise.
· Line 76: the connection between the sentence describing the roller mill qualities and the sentence describing the hammer mill qualities is missing. Please revise.
· Line 93: again, the connection between the combination of hammer and roller mill and chahua chicken #2 is missing – please revise.
Reviewer 3 Report
The objective of the work was to compare the effects of grinding method on nutrient digestibility of corn in chickens. There are thousands of papers that have been published related to grinding method. The authors need to emphasize what is novel in this work. There are some suggestions for improvement of the paper detailed below. The work is in need of some grammatical editing.
One of the main issues is that the treatments are not clear. Line 25 (and elsewhere) implies that a roller mill, hammer mill and 2 stage grinding was used. This suggests 3 treatments. However, Table 2 shows 4 groups and my interpretation is that both roller and hammer mills were used in each of the 4 groups. This must be clarified.
What was the diet fed to the birds during the experiment? Surely the birds were not only fed corn for the 19 weeks of the study. During the testing periods (week 12 and 19) were the birds only fed corn or was it supplemented with vitamins and minerals?
Title If this work was published exclusively in China the term “native” chicken would be sufficient. However, since readers will be international, a more specific description of the birds is needed in the title.
Line 27 Change “native” to “Chahua”.
Line 94 Additional description of the bird should be provided. For example, other publications indicate this “is considered to be a primitive type of chicken that exhibits many phenotypes and behaviors similar to those of the red jungle chicken.” It might be helpful to cite other publications that compare this line to more conventional birds. Commercial chickens typically reach 2 kg BW in 5 weeks as compared to 120 weeks here. So this line is extremely different that what most of the world consumes for meat.
Line 121 Clarify whether the same batch of “corn” was used at both 12 and 19 weeks and for all groups.
Line 121 A more specific description of “corn” is needed. What was the variety, where was it grown, etc.? Corn with 10% crude protein (Table 1) is not common – most is closer to 8% or less.
Line 138 Indicate how many replicates were used for grinding here and in Table 3. Were multiple samples obtained from the same grind or were there multiple grinds for each condition? In other words, is the data in Table 3 measuring the variation in samples of the same batch or different batches?
Line 151 Were the same birds used for week 12 and 19? Were birds given the same grind process at week 12 and 19? What were the bird density in the cages?
Line 159 Clarify that only corn was fed and that it was not supplemented with vitamins and minerals. What was the diet fed from week 0-12, 13-19?
Line 160 How was feed spillage separated from excreta?
Line 161 What was the ratio of excreta to HCl used?
Line 186 Again, it is not clear what the groups are. Looking at Table 3, group M1 appears to have both roller and hammer mill processing. The authors MUST clarify what the groups were.
Line 208 What are the units for dgw and Sgw? I assume it is microns for dgw.
Line 211 Table 4 – clarify that the GE, protein and amino acid values are for the excreta. Was there any statistical analysis of this data?
Line 220 Should be “Discussion”.
The text is in need of minor grammatical editing throughout.
